# ST11 Carbapenem-Resistant *Klebsiella pneumoniae* Clone Harboring *bla*_NDM_ Replaced a *bla*_KPC_ Clone in a Tertiary Hospital in China

**DOI:** 10.3390/antibiotics11101373

**Published:** 2022-10-07

**Authors:** Qiaoyan Duan, Qi Wang, Shijun Sun, Qiaozhen Cui, Qi Ding, Ruobing Wang, Hui Wang

**Affiliations:** 1Department of Clinical Laboratory, Shanxi Provincial People’s Hospital, Taiyuan 030012, China; 2Department of Clinical Laboratory, Peking University People’s Hospital, Beijing 100044, China

**Keywords:** carbapenem-resistant Enterobacterales, *Klebsiella pneumoniae*, carbapenemase, KPC-2, NDM, whole-genome sequencing

## Abstract

The nosocomial spread of carbapenem-resistant Enterobacterales (CRE) is extremely common, resulting in severe burdens on healthcare systems. In particular, the high-risk *Klebsiella pneumoniae* ST11 strain has a wide endemic area in China. The current study describes the results of continuous monitoring of CRE genotypes and phenotypes in a tertiary hospital in North China from 2012 to 2020. A total of 160 isolates were collected, including 109 *Klebsiella. pneumoniae* (68.13%), 29 *Escherichia coli* (26.60%), 12 *Enterobacter cloacae* (7.50%), and 10 other strains (6.25%). A total of 149 carbapenemase genes were detected, of which *bla*_KPC__-2_ (51.0%) was the most common, followed by *bla*_NDM__-1_ (22.82%), and *bla*_NDM__-5_ (23.49%). Based on multi-locus sequence typing, the ST11 strain (66.1%) dominates *K. pneumoniae*, followed by ST15 (13.8%). Interestingly, the proportion of *bla*_NDM_ (22.2%, 16/72) in ST11 *K. pneumoniae* was significantly increased in 2018–2019. Hence, whole-genome sequencing was performed on ST11 *K. pneumoniae.* Growth curves and in vitro competition experiments showed that *K. pneumoniae* carrying *bla*_NDM_ exhibited a stronger growth rate (*p* < 0.001) and competition index (*p* < 0.001) than *K. pneumoniae* carrying *bla*_KPC_. Moreover, *K. pneumoniae* carrying *bla*_NDM_ had a stronger biofilm-forming ability than *K. pneumoniae* carrying *bla*_KPC_ (*t* = 6.578; *p* < 0.001). *K. pneumoniae* carrying *bla*_KPC_ exhibited increased defense against bactericidal activity than *K. pneumoniae* carrying *bla*_NDM_. Thus, ST11 *K. pneumoniae* carrying *bla*_NDM_ has strong adaptability and can locally replace *K. pneumoniae* carrying *bla*_KPC_ to become an epidemic strain. Based on these findings, infection control and preventive measures should focus on the high-risk ST11-*K. pneumoniae* strain.

## 1. Introduction

Enterobacterales are the most common pathogenic gram-negative bacteria responsible for causing myriad community-acquired and healthcare-acquired infections [1,2]. In particular, carbapenem-resistant Enterobacterales (CRE) have been reported worldwide, including in China [3,4,5], and represent a significant public health challenge. In 2017, the World Health Organization (WHO) released a list of drug-resistant bacteria that pose a significant threat to human health, including CRE, and highlighted the urgent need to develop new antibiotics [6].

Carbapenem resistance in Enterobacterales can arise through several molecular mechanisms, the most common of which include carbapenemase production. According to the Ambler classification [7,8,9], class A (mainly *Klebsiella pneumoniae* carbapenemase [KPC]), class B (mainly New Delhi metallo-β-lactamase [NDM], imipenemase [IMP], and Verona integron-encoded metallo-β-lactamase [VIM]), and class D (mainly oxacillinase [OXA]-48-like) are the primary classes of CRE carbapenemases. The second most common resistance mechanism includes the high production of extended-spectrum β-lactamases (ESBL) or AmpC cephalosporinase, combined with the decreased permeability of the outer membrane proteins. Moreover, active efflux pumps reportedly lead to carbapenemase resistance [10,11]. 

*K. pneumoniae* is a gram-negative bacillus belonging to the order Enterobacterales which can cause a wide range of infections in community and healthcare settings, leading to increased morbidity and mortality [12,13]. In clinical settings, *K. pneumoniae* and *Escherichia coli* represent the most commonly isolated carbapenemase-producing species. KPC-2 and NDM-1 are the predominant carbapenemases in carbapenem-resistant *K. pneumoniae* (CRKP) and *Escherichia coli*, respectively [4,5]. Moreover, sequence type 11 (ST11) is the dominant ST in *K. pneumoniae* within Asia, while ST258 dominates Europe and America; these strains are closely related [3].

According to reports by the China Antimicrobial Resistance Surveillance System, the detection rate of CRE in China is increasing (http://www.carss.cn/Report/Details?aId=808, accessed on 1 December 2021). The incidence of CRKP in China increased from 5.6% in 2014 to 10.9% in 2020. Importantly, CRE have geographical and temporal variations in different global regions. In 2020, the incidence of CRKP in China also varied greatly in different regions. For example, the region with the highest incidence was Henan Province, with 30.2%, Shanghai with 24.7%, and Beijing with 25.2%. Meanwhile, the incidence of CRKP in Inner Mongolia, Guangxi, Shanxi, Heilongjiang, Hainan, Gansu, Jilin, Ningxia, Qinghai, and Tibet was all below 5%. Differing drug resistance rates and detection rates are closely related to the prevention and control of nosocomial infections and the rational use of antibiotics. Whole genome sequencing (WGS) has been widely used to control and prevent multidrug-resistant organism infections [14,15]. This study screened 160 CRE isolates from a tertiary hospital in Shanxi province, China, from 2012 to 2020. Whole-genome sequencing data were combined with epidemiological data to analyze the CRKP genetic evolution and population migration to provide important insights for developing effective infection prevention and control measures.

## 2. Materials and Methods

### 2.1. Description of Bacterial Isolates

This was a single-center, nine-year long, continuous molecular epidemiological monitoring study of CRE strains. From 1 January 2012 to 31 July 2020, a total of 160 unique clinical CRE isolates were collected from Shanxi Provincial People’s Hospital (China), a large 2000-bed tertiary hospital in Shanxi province. The level of resistance of Enterobacterales to any carbapenem (imipenem, meropenem, or ertapenem) was determined by the agar dilution method [16]. All isolates were reidentified by matrix-assisted laser desorption ionization time-of-flight mass spectrometry (Bruker Daltonics Inc., Billerica, MA, USA) at Peking University People’s Hospital and were stored at −80 °C for antimicrobial susceptibility testing and investigation of resistance mechanisms.

### 2.2. Carbapenemase-Producing Phenotypic Tests

The modified carbapenem inactivation method (mCIM) and EDTA-modified carbapenem inactivation method (eCIM) were used to test carbapenemase production according to Clinical and Laboratory Standards Institute (CLSI) M100 S31 guidelines [17]. Briefly, the mCIM procedure for each isolate to be tested comprised emulsification of a 1-μL loopful of bacteria from an overnight blood agar in 2 mL of tryptic soy broth (TSB) (OXOID, Hampshire, UK). After vortexing for 10 s, a 10-μg meropenem disk (OXOID, Hampshire, UK) was added to the broth, which was incubated at 35 °C in ambient air for 4 h. Subsequently, a Mueller–Hinton agar (MHA) (OXOID, Hampshire, England) plate was inoculated with *E. coli* ATCC^®^ 25922 in a 0.5 McFarland suspension. The meropenem disk was removed from the TSB using a 10 μL loop. Then, the meropenem discs were affixed to the MHA inoculated with *E. coli* ATCC^®^ 25922. We made sure to minimize the TSB broth’s liquid residue during the above process. The MHA plates were inverted and incubated at 35 °C in ambient air for 18–24 h. For each isolate, a second 2 mL TSB tube was labeled for the eCIM test. Next, 20 μL of the 0.5 M EDTA was added to the 2 mL TSB tube to obtain a final concentration of 5 mM EDTA, after which the steps for the mCIM procedure were repeated. The mCIM and eCIM tubes were assessed in parallel by placing the meropenem disks from the mCIM and eCIM tubes on the same MHA plate inoculated with 25922 indicator strain. *K. pneumoniae* ATCC^®^ BAA 1705 and *K. pneumoniae* ATCC^®^ BAA 1706 served as the positive and negative controls, respectively. As previously described, PCR detection of common carbapenemase genes (*bla*_KPC_, *bla*_NDM_, and *bla*_IMP_) was performed for mCIM-positive isolates [5]. After Sanger sequenced the amplification results, we compared the sequences on the http://www.bldb.eu/Enzymes.php (accessed on 1 December 2021) website to identify each carbapenemase [18].

### 2.3. Antimicrobial Susceptibility Testing

The agar dilution method determined the minimum inhibitory concentrations (MICs) of all the antimicrobial agents except colistin. MICs of colistin were determined by the broth microdilution method. The results were interpreted according to CLSI M100 S30 guidelines [17]. The breakpoint of tigecycline for Enterobacterales was based on the US Food and Drug Administration standard (≤2 μg/mL is susceptible, ≥8 μg/mL is resistant). *Escherichia coli* ATCC^®^ 25922 and *Pseudomonas aeruginosa* ATCC^®^ 27853 were used as quality control isolates. The tested antimicrobial agents included imipenem, meropenem, ertapenem, colistin, tigecycline, ceftazidime, cefepime, cefoperazone-sulbactam, amikacin, aztreonam, piperacillin-tazobactam, ciprofloxacin, fosfomycin, levofloxacin, and minocycline.

### 2.4. Multi-Locus Sequence Typing (MLST)

According to the Institute Pasteur MLST procedure (https://bigsdb.pasteur.fr/, accessed on 1 December 2021), *K. pneumoniae* (https://bigsdb.pasteur.fr/klebsiella/primers-used/, accessed on 1 December 2021) and *Escherichia coli* (https://bigsdb.pasteur.fr/ecoli/primers-used/, accessed on 1 December 2021) MLST methods were performed to amplify and sequence seven housekeeping genes. After uploading and comparing the sequencing results, the allele number and STs were obtained. Based on the MLST allele profile, the phylogenetic results were visualized using the PHYLOViZ online application (http://www.phyloviz.net/, accessed on 1 December 2021).

### 2.5. WGS and Phylogenetic Analysis

WGS and evolutionary correlation analysis were performed for *K. pneumoniae* ST11 strains isolated throughout the study. Total genomic DNA was extracted using the TIANamp Bacteria DNA Kit DP302 (Tiangen Biotech, Beijing, China), followed by genomic DNA sequencing using the Illumina HiSeq X Ten platform (Illumina Inc., San Diego, CA, USA), which produced 150 bp paired-end reads and at least 100-fold coverage of raw reads. The short-read sequence was assembled *de novo* using SPAdes v3.10.0. All draft genome sequences were deposited into the National Center for Biotechnology Information’s genome database and organized under BioProject ID PRJNA867316. All resistance genes were detected by the Resfinder (https://cge.cbs.dtu.dk/services/ResFinder/, accessed on 1 December 2021) and basic local alignment search tool. K-locus (polysaccharide capsule) typing was identified with the Kaptive software [19] using the whole-genome sequences. The core genome’s maximum likelihood phylogenetic analysis was performed using RAxML (version v7.2.8, Alexandros Stamatakis, Karlsruhe, Germany). Visualization of the evolutionary tree was performed using iTOL (https://itol.embl.de/, accessed on 1 December 2021).

### 2.6. Growth Assay and In Vitro Competition Experiment

To compare the fitness costs of *K. pneumoniae* carrying different carbapenemase genes, isolates carrying *bla*_KPC_ (C5343 [K2], C7673 [K3], and C7676 [K1]) and *bla*_NDM_ (C7662 [N2], C7664 [N3], and C7674 [N1]) were subjected to growth curve assay and in vitro competition growth experiments, as previously described [20]. Briefly, isolates were cultured overnight in Luria-Bertani (LB) broth diluted to an OD_600_ of 0.01 and grown at 37 °C with shaking (200 rpm). Triplicate experiments were run for each isolate. The culture cell density was determined every 0.5 h by measuring the OD at 600 nm (Thermo Fisher Scientific, Shanghai, China).

In the in vitro competition assays, six isolates were separately cultured overnight in LB broth at 37 °C. The bacteria were diluted to 0.5 × 10^6^ colony-forming units (CFU)/mL, and equal volumes of K1/N1, K2/N2, and K3/N3 were combined and cultured at 37 °C with shaking (200 rpm). NDM-producing *K. pneumoniae* can grow on plates containing ceftazidime-avibactam; however, KPC-producing *K. pneumoniae* cannot. At 0, 4, 8, 12, and 24 h, aliquots of the mixed bacteria were diluted with 0.9% saline solution and plated on LB agar plates with or without ceftazidime-avibactam (32 μg/mL). CFU were counted after 18 h of incubation at 37 °C. The competitive index (CI) was determined as follows: CI = (N1/K1) (Inoculated N1/Inoculated K1), as previously described [21].

### 2.7. Serum-Killing Assay

Serum-killing assays were conducted to determine virulence in vitro as previously described [22]. Briefly, human blood sera were obtained from ten healthy individuals. *K. pneumoniae* isolates were inoculated into LB broth medium and incubated at 37 °C with shaking until the logarithmic phase was reached (OD_600_ = 0.6). An inoculum of 50 µL diluted culture (containing 1 × 10^6^ CFU) was added into 150 µL of pooled human sera contained in a 10 × 75 mm Falcon polypropylene tube (BD Biosciences, Franklin Lakes, NJ, USA) and incubated at 37 °C with shaking. Viable counts were checked at 0, 1, 2, and 3 h, and aliquots of the bacteria were diluted with 0.9% saline solution and plated on LB agar plates. CFU were counted after 18 h of incubation at 37 °C. Each strain was tested at least three times. The mean results were expressed as inoculation percentages, and six grades were assigned to classify the strains as described previously [23]. *K. pneumoniae* K2044 and *K. pneumoniae* ATCC^®^ 13883, which were identified as serum killing sensitive (grade 1) and resistant (grade 5) in our previous study [23], were used as negative and positive controls, respectively.

### 2.8. Biofilm Formation Assay

Biofilm production was determined as described recently [22]. Cultures that were shaken overnight were diluted 1:1000 in LB broth and a total volume of 200 µL was inoculated into wells of an untreated 96-well microplate, with six wells per strain. After 24 h incubation at 37 °C, the LB broth was removed, the wells were washed four times with deionized water and 200 µL of 0.1% crystal violet was added. After a 15 min incubation, crystal violet was removed, and the wells were washed six times with deionized water. Subsequently, 200 µL of 30% acetic acid was added, and the plate was incubated for 10 min at room temperature before determining the OD 590 with Varioskan Flash (Thermo Fisher Scientific, Shanghai, China).

### 2.9. Extraction and Quantification of Capsules

The capsule was extracted, and uronic acids were quantified to assess capsule production. Overnight shaking cultures grown in LB were adjusted to an OD_600_ of 0.6 with LB broth. The capsule was extracted using the modified protocol described previously [24]. Briefly, 500 µL cultures were mixed with 100 µL of 1% Zwittergent 3–14 detergent in 100 mM citric acid (pH 2.0) and heated for 30 min at 50 °C with occasional vortexing. Then, the above mix was centrifuged at 16,000× *g* for five minutes. After centrifuging, 250 µL of the supernatant was transferred to a new tube and precipitated with 1 mL of 80% ethanol at four °C for 30 min. The precipitate was centrifuged at 16,000× *g* for 5 min and dried, and uptake was performed in 100 µL of deionized water and 12.5 mM sodium tetraborate-sulfuric acid solution. The samples were mixed and boiled at 100 °C for 15 min. A series of tubes containing two-fold serial dilutions of glucuronolactone were standard. The samples were cooled to room temperature for approximately 30 min before adding 10 µL of 0.15% m-hydroxybiphenyl and further incubation for 15 min to generate the chromophore. The OD was determined at 520 nm. Uronic acids were quantified by comparing the OD with the glucuronolactone standard curve.

### 2.10. Statistical Analysis

Statistical analysis was performed with the GraphPad Prism version 9 (GraphPad Software, San Diego, CA, USA) using one-way analysis of variance followed by Tukey–Kramer tests. Statistical significance was set at *p* < 0.05. Data from the antibiotic susceptibility tests were analyzed using WHONET 5.6 software (WHO, Geneva, Switzerland). 

### 2.11. Ethics Statement

The Research Ethics Committee of Shanxi Provincial People’s Hospital approved this study. Medical records and patient information were retrospectively reviewed and collected. Informed consent was not required because the medical records and patient information were anonymously reviewed and collected in this observational study.

## 3. Results

### 3.1. Distribution of Isolates from 2012 to 2020

Figure 1 shows the number of CRE isolates per year. Respiratory tract infections were the predominant source of infection, accounting for 46.8% (75/160), followed by urinary tract (19.3%; 31/160) and bloodstream (15.6%; 25/160). Among the 160 CRE strains, *K. pneumoniae* accounted for the highest proportion (68.1%), followed by *Escherichia coli* (18.1%). In this study, CRE strains were mainly isolated in the neurosurgery (25%; 40/160), intensive care unit (ICU; 16.8%; 27/160), neurology (10%; 16/160), and respiratory (6.9%; 11/160) wards.

### 3.2. Antimicrobial Susceptibility Testing

Antimicrobial susceptibility testing results are summarized in Table 1. Of all the antimicrobial agents tested, colistin had the highest susceptibility (98.8%), followed by tigecycline (90%), amikacin (65%), and fosfomycin (52.2%), while susceptibility to the remaining antimicrobial agents was <50%. In particular, the tested isolates had low susceptibility to imipenem (8.1%), meropenem (9.4%), and ertapenem (1.9%). Apparent interspecies variations in susceptibility were detected. For example, *Escherichia coli* was less susceptible to aztreonam than other species (41.4% vs. 14.4%, *p* < 0.05). The susceptibility rate of *K. oxytoca* to minocycline was 87.5%, compared with 8.3% for *Enterobacter cloacae*. Moreover, only 5.5% of *K. pneumoniae* isolates were susceptible to ciprofloxacin, compared with 41% of *Enterobacter cloacae* and *K. oxytoca* isolates.

### 3.3. Distribution of STs and Carbapenemases

A total of 15 STs were identified in 109 CRKP isolates (Table 2). ST11 was the most prevalent (66.1%; 72/109) sequence type (ST), followed by ST15 (13.8%), ST48 (8.33%), ST1786, ST357, and ST616, which is rarely reported in China [5,25]. Seventy-one *K. pneumoniae* strains carrying the *bla*_KPC-2_ gene were classified into five ST_S_, including ST11 (56; 78.87%), ST15 (7; 9.86%), ST48 (6; 8.45%), ST25, and ST86 (1; 1.41%). Moreover, nearly all ST11 isolates produced carbapenemases (98.61%, 71/72), particularly *bla*_KPC-2_ (77.5%) and *bla*_NDM-1_ (22.5%). *bla*_NDM-5_ and *bla*_NDM-1_ were the most common carbapenemase genes in *Escherichia coli* (96%; 24/25) and *Enterobacter cloacae* (58.3%; 7/12), respectively. Figure 2 shows the minimum spanning tree based on MLST of *K. pneumoniae* collected from 2012 to 2016 in this study.

In this study, 149 (93.13%) isolates produced carbapenemases. According to the PCR results, 76 (51.0%), 35 (23.49%), 34 (22.82%), two (1.34%), and one (0.67%) isolates harbored *bla*_KPC-2_, *bla*_NDM-1_, *bla*_NDM-5_, *bla*_NDM-4_, and *bla*_IMP-4_, respectively (Table 2). Uncommon genes were not detected. However, one *K. oxytoca* co-harboring *bla*_KPC-2_ and *bla*_NDM-1_ was detected.

### 3.4. Phylogenetic Analysis of ST11 K. pneumoniae Isolates

We performed WGS of all ST11 *K. pneumoniae* isolates. The phylogenetic tree revealed that ST11 isolates harboring *bla*_KPC-2_ could be assigned to ten clades (Figure 3). CRKP-ST11-*bla*_NDM_ isolates were in the same branch and had no noticeable differences, while CRKP-ST11-*bla*_KPC_ isolates were in a separate independent branch, indicating that CRKP-ST11-*bla*_NDM_ may be a novel type. Interestingly, the *bla*_NDM_-carrying strains replaced the *bla*_KPC_-carrying strains in prevalence over time, especially from 2018 to 2019, when CRKP-ST11-*bla*_NDM_ was first isolated. According to the *wzi* typing results, four different capsular serotypes were identified. K47 was the most common serotype (37.6%), followed by K111 (14.68%), K64 (11%), and K23 (1.8%). Seventy-five percent of the K64 isolates harbored *rmpA* and *rmpA2* virulence genes, and 76.38% of the ST11 isolates harbored *ybt9*ICE*Kp3*. Furthermore, 71 *K. pneumoniae* isolates harbored genes encoding extended-spectrum β-lactamases (mainly CTX-M-65 and CTX-M-15).

### 3.5. Growth Assay and In Vitro Competition Experiment

In growth assay, K1, K2, and K3 were significantly slower than N1, N2, and N3 isolates at various time points from the fifth hour to the thirteenth hour (*p* < 0.05; Figure 4; Appendix A). In vitro competition experiments revealed that *K. pneumoniae*-*bla*_NDM_ overgrew *K. pneumoniae*-*bla*_KPC_ with a mean confidence interval (CI) of 1.60 ± 0.17, 2.88 ± 0.33, 3.66 ± 0.69, and 5.44 ± 1.11 at 4, 8, 12 and 24 h, respectively (Appendix A). Figure 5 shows three groups of competitive growth in vitro: K1 vs. N1, K2 vs. N2, and K3 vs. N3 isolates cultured at 37 °C in LB broth (Appendix A). Data points represent three independent experiments’ mean (±standard deviation). Among the pairwise comparison of growth curves, K1 vs. N1, K2 vs. N2, and K3 vs. N3 showed significant (*p* < 0.05) differences. 

### 3.6. Serum Killing Assay

Different *K. pneumoniae* strains were compared in terms of their ability to survive the bactericidal activity of human serum (Table 3). *K. pneumoniae*-*bla*_KPC_ had intermediate sensitivity to serum killing (grade 3–6), whereas *K. pneumoniae*-*bla*_NDM_ was sensitive (grade 2), thus suggesting that *K. pneumoniae*-*bla*_KPC_ exhibited a higher defense against the bactericidal activity of sera than *K. pneumoniae*-*bla*_NDM_. 

### 3.7. Biofilm Formation Assay 

The biofilm formation capacities of NDM-1-positive and KPC-2-positive *K. pneumoniae* isolates measured at A590 were 0.096 ± 0.037 and 0.133 ± 0.039, respectively (*t* = 6.578; *p* < 0.0001; Figure 6). The biofilm formation capacity of *K. pneumoniae* carrying NDM-1 was significantly higher than that of *K. pneumoniae* carrying KPC-1.

### 3.8. Quantification of Capsule Production

Uronic acids were quantified to assess capsule production. We found that capsule production was lower in the *K. pneumoniae*-*bla*_NDM_ isolates. However, the two groups had no significant difference (Appendix A).

## 4. Discussion

CRE infection has become a significant global public health issue due to the widespread use and inappropriate application of carbapenems. In our hospital in Shanxi province, China, the prevalence of CRKP increased from 0.8% in 2012 to 5.73% in 2019, thus reflecting an upward trend and revealing that *K. pneumoniae* is the primary domestic CRE strain [5,26]. 

Respiratory tract infections were the most common source of isolation, which agrees with the results reported by Chen et al. in China [27]. In this study, CRE strains were mainly isolated in the neurosurgery (25%), intensive care unit (ICU; 16.8%), neurology (10%), and respiratory (6.9%) wards. This was likely due to the common use of mechanical ventilation in these departments, which is closely related to the intensity of antibiotic use, invasive operations, and low immunities of patients [28]. The ICU is the department with the most cases of CRE infection. Importantly, the in-hospital transport of patients in the ICU and other departments increases the risk of CRE in-hospital transmission [29]. Therefore, infection control in these departments may be beneficial in preventing the spread of CRE.

The CRE strains used in this study showed extensive drug resistance, consistent with China’s surveillance data [26,30]. All isolates exhibited resistance to carbapenems and cephalosporins. There were apparent interspecies variations in susceptibility; for example, *Escherichia coli* was less susceptible to aztreonam than other species. 

Consistent with changing domestic trends [31,32], our research revealed that 93.13% of CRE strains produced carbapenemases, among which KPC was predominant in *K. pneumoniae* isolates, and NDM-5 was predominant in *Escherichia coli* isolates; NDM-1 was the most common type in *Enterobacter cloacae*. Interestingly, studies have shown that the resistance mechanisms of CRE clinical strains isolated from children or adults, or from different geographical distributions differ. In particular, KPC-2-producers are widespread in adult patients and can be spread horizontally through plasmids, thus causing potential outbreaks. Notably, CRE strains that co-produce NDM and KPC carry many resistance genes, making them highly resistant to the most commonly used antibiotics [33].

Seventy-one *K. pneumoniae* strains carrying the *bla*_KPC-2_ gene were classified into five STs, including ST11, ST15, ST48, ST25, and ST86. MLST showed subtype diversity and more heterogeneous clonal backgrounds of CRKP isolates related to nosocomial pneumonia. All ST48 isolates were detected in the neurosurgery ward, and may have resulted from the nosocomial transmission. Moreover, the proportion of *bla*_NDM_ (22.2%, 16/72) in ST11 *K. pneumoniae* significantly increased in 2018–2019. This finding may indicate local epidemics, suggesting that more attention should be paid to the spread of *bla*_NDM_ in the ST11 strain. 

K47 was the most common *K. pneumoniae* serotype in our study. Similarly, in a study by Zhou et al., K64 was the most common serotype among 16 carbapenem-resistant hvKP isolates [34]. This single-center study showed that K64 had a higher virulence gene carrier rate than K47. The hypermucoviscous phenotype of hypervirulent *K. pneumoniae* (hvKP) is typically due to the increased production of capsular polysaccharides and specific virulence genes, such as *rmpA* and *rmpA2*. The emergence of and rapid increase in hvKP strains in recent years, especially carbapenemase-producing hvKP-related infections in immunocompromised patients, are worrisome and pose a severe threat to patients [25,35,36,37].

Our in vitro competition study revealed that *K. pneumoniae*-*bla*_KPC_ overgrew *K. pneumoniae*-*bla*_NDM_. Meanwhile, the proportion of NDM-1-positive *K. pneumoniae* isolates with biofilm formation capacity was higher than KPC-2-positive *K. pneumoniae* isolates. However, *K. pneumoniae*-*bla*_KPC_ exhibited greater defense against bactericidal activity than *K. pneumoniae*-*bla*_NDM_. The virulence evaluation criterion was the hypermucoviscous phenotype, biofilm formation ability, and serum resistance [38]. In this study, virulence factors were not detected in the NDM-1-positive *K. pneumoniae* group, which is consistent with the drawing experiment and the K111 capsule serotypes. Fuursted et al. reported that *K. pneumoniae* carrying NDM-1 have an intrinsic virulence potential [39]. Interestingly, Montanari claims that bacteria lose certain virulence genes to obtain resistance genes for optimal adaptability [40]. Therefore, virulence influences the decrease in biofilm formation and growth rate. Indeed, there are several cases of carbapenem-resistant hvKP carrying *bla*_NDM-1_ in China [41,42].

Through our research, we came to the following conclusions. ST11 *K. pneumoniae* carrying *bla*_NDM_ has strong adaptability and can locally replace *K. pneumoniae* carrying *bla*_KPC_ to become an epidemic strain. Infection control and preventive measures should focus on high-risk ST11-*K. pneumoniae* strains.

## Figures and Tables

**Figure 1 antibiotics-11-01373-f001:**
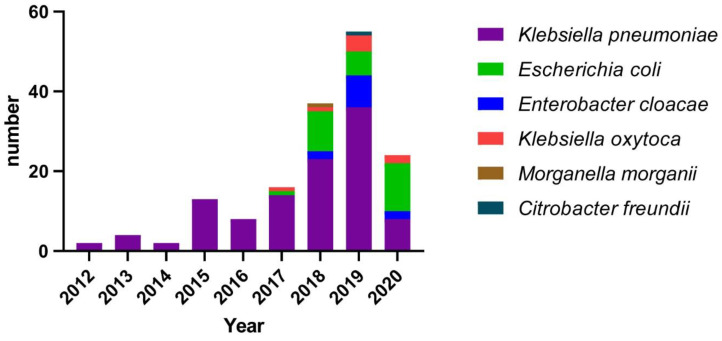
Species distribution of all carbapenem-resistant Enterobacterales strains from 2012 to 2020 in this study.

**Figure 2 antibiotics-11-01373-f002:**
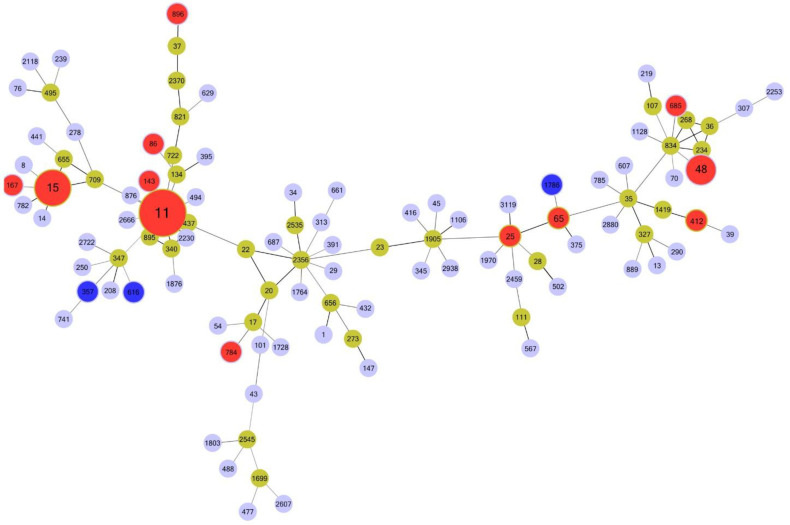
Minimum spanning tree of multi-locus sequence typing (MLST) of *Klebsiella pneumoniae* collected from 2012 to 2020. Each circle (node) represents multiple identical sequences. Circle diameter in red is based on the number of isolates on a log scale. Red and blue represent *Klebsiella pneumoniae* ST type in this study. Blue is ST type not reported in the CRE-Network study [5].

**Figure 3 antibiotics-11-01373-f003:**
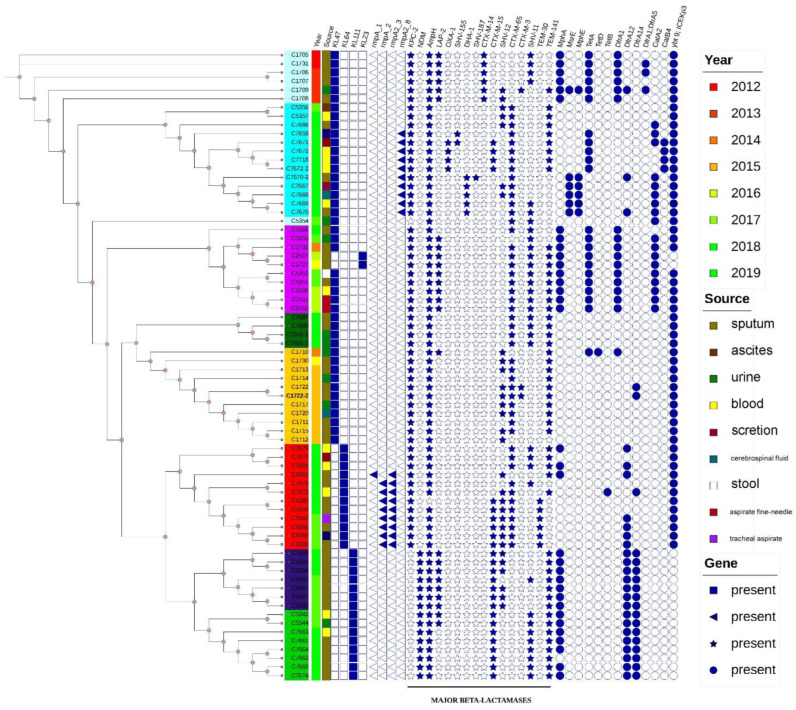
Phylogenetic tree of the core genome for all sequence type 11 *Klebsiella pneumoniae* strains.

**Figure 4 antibiotics-11-01373-f004:**
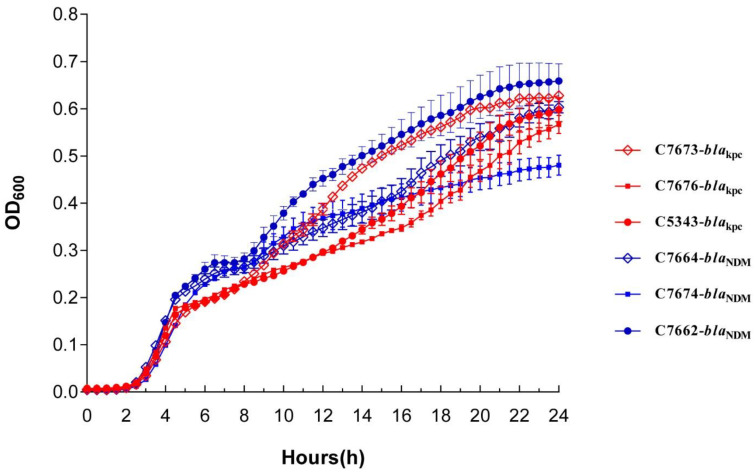
Growth curve for six strains of *Klebsiella pneumoniae* (C7676-*bla*_KPC_, C5343-*bla*_KPC_, C7673-*bla*_KPC_, C7674-*bla*_NDM_, C7662-*bla*_NDM_, and C7664-*bla*_NDM_). Error bars represent standard deviation.

**Figure 5 antibiotics-11-01373-f005:**
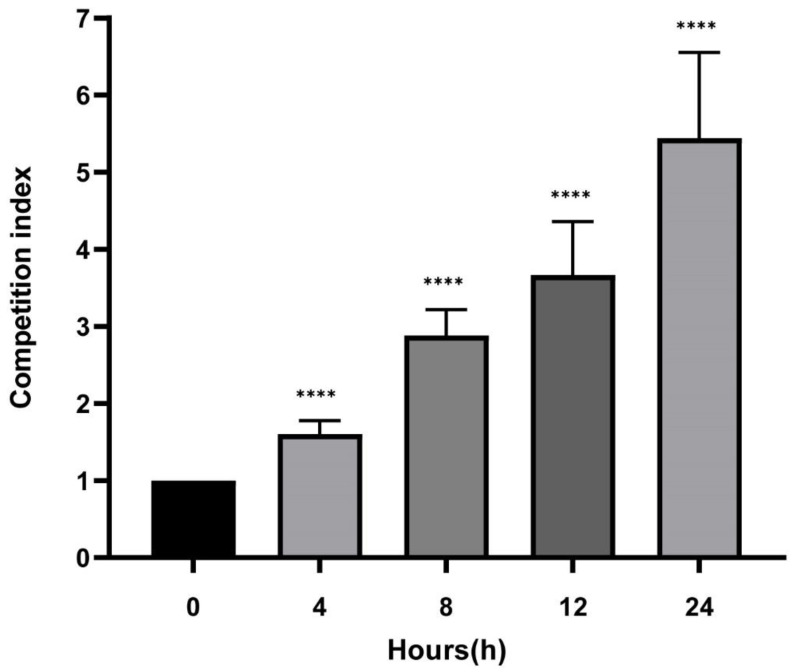
In vitro bacterial competition experiment of three group. N1/K1, N2/K2, N3/K3 mixed inoculum were grown in vitro; the competition index is shown from 0 to 24 h. **** *p* < 0.001 between 0 and time of inoculation. Error bars represent standard deviation.

**Figure 6 antibiotics-11-01373-f006:**
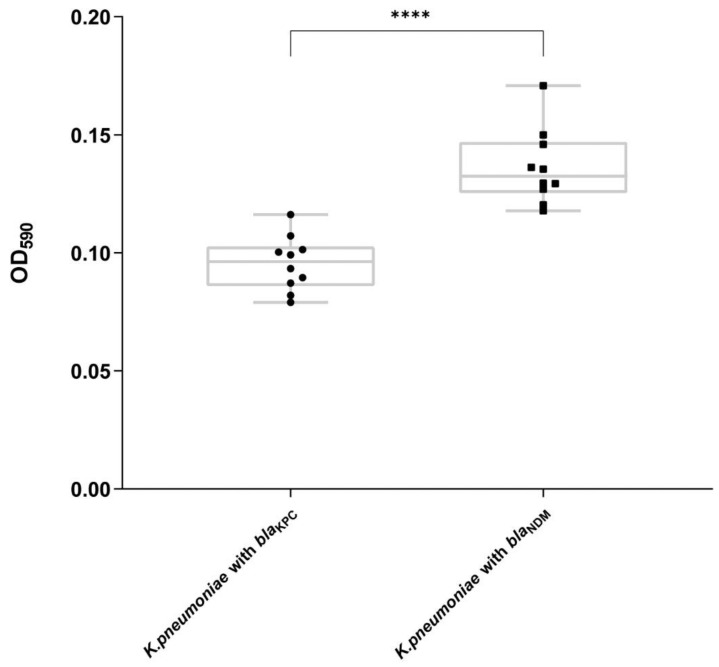
Biofilm formation. Circles indicate the ten *Klebsiella pneumoniae* carbapenemase 2 (KPC-2)-positive *K. pneumoniae* isolates; squares indicate the ten New Delhi metallo-β-lactamase (NDM)-1-positive *K. pneumoniae* isolates. **** means statistical significance (*p* < 0.01) between *K. pneumoniae*-*bla*_KPC_ and *K. pneumoniae*-*bla*_NDM_. Error bars represent standard deviation.

**Table 1 antibiotics-11-01373-t001:** Antimicrobial Susceptibility Testing Results of CRE Isolates.

Antibiotic Name	All Isolates (*n* = 160)				*Klebsiella pneumoniae* (*n* = 109)				*Escherichia coli* (*n* = 29)			
	%R	%S	MIC_50_	MIC_90_	%R	%S	MIC_50_	MIC_90_	%R	%S	MIC_50_	MIC_90_
Amikacin	35	65	4	>256	42.2	57.8	4	>256	24.1	75.9	4	256
Aztreonam	82.5	14.4	128	>256	89.9	6.4	256	>256	55.2	41.4	16	>256
Cefepime	94.4	1.2	128	>256	95.4	1.8	128	256	96.6	0	256	>256
Cefoperazone-sulbactam	98.1	0.6	>256	>256	97.2	0.9	>256	>256	100	0	>256	>256
Ceftazidime	97.5	1.9	>256	>256	96.3	2.8	256	>256	100	0	>256	>256
Ciprofloxacin	86.9	10.6	64	128	93.6	5.5	64	128	89.7	6.9	64	128
Colistin	1.2	98.8	0.25	0.5	0	100	0.25	0.5	3.4	96.6	0.25	0.25
Ertapenem	96.9	1.9	32	256	97.2	1.8	64	256	93.1	3.4	32	64
Fosfomycin	41.5	52.2	64	>256	55.6	36.1	256	>256	13.8	86.2	4	>256
Imipenem	87.5	8.1	16	64	89.9	6.4	16	64	79.3	10.3	4	16
Levofloxacin	85	14.4	32	128	90.8	8.3	32	128	89.7	10.3	32	64
Meropenem	85	9.4	16	128	90.8	5.5	32	128	72.4	20.7	4	16
Minocycline	41.9	49.4	8	32	38.5	53.2	4	32	48.3	41.4	8	128
Piperacillin-tazobactam	95	2.5	>256	>256	96.3	2.8	>256	>256	93.1	3.4	256	>256
Tigecycline	1.9	90	0.5	2	0.9	88.1	0.5	4	0	100	0.25	0.5

Abbreviations: CRE, carbapenem-resistant Enterobacterales; MIC_50/90_, 50%/90% minimum inhibitory concentration; %R, percentage of resistant isolates; and %S, percentage of susceptible isolates.

**Table 2 antibiotics-11-01373-t002:** Carbapenemase Distribution Determined by CRE MLST.

Organism	ST	No.	IMP-4 *n* (%)	KPC-2 *n* (%)	NDM-1 *n* (%)	NDM-4 *n* (%)	NDM-5 *n* (%)
*Klebsiella pneumoniae* (*n* = 102)	ST11	71		55 (77.5)	16 (22.5)		
	ST15	14		7 (50)	5 (35.7)	2 (14.3)	
	ST25	1		1 (100)			
	ST48	6		6 (100)			
	ST86	1		1 (100)			
	ST143	1					1 (100)
	ST357	1					1 (100)
	ST412	1			1 (100)		
	ST685	1					1 (100)
	ST784	1	1 (100)				
	ST896	1			1 (100)		
	Other STs	3		1 (33.3)	1 (33.3)		1 (33.3)
*Escherichia coli* (*n* = 25)	ST410	7					7 (100)
	ST167	4					4 (100)
	ST617	3					3 (100)
	ST10	2					2 (100)
	ST354	2					2 (100)
	ST744	2					2 (100)
	ST46	1					1 (100)
	ST48	1					1 (100)
	ST69	1			1 (100)		
	ST648	1					1 (100)
	ST10565	1					1 (100)
*Enterobacter cloacae* (*n* = 12)	ST175	5			5 (100)		
	ST97	2					2 (100)
	ST171	2					2 (100)
	ST116	1					1 (100)
	ST177	1			1 (100)		
	ST231	1			1 (100)		

Abbreviations: CRE, carbapenem-resistant Enterobacteriaceae; IMP, imipenemase; KPC, *Klebsiella pneumoniae* carbapenemase; MLST, multi-locus sequence typing; NDM, New Delhi metallo-β-lactamase; and ST, sequence type.

**Table 3 antibiotics-11-01373-t003:** Serum resistance analysis based on serum killing assay.

Isolate	Hour (h)			Grade	Interpretive
	1 (%)	2 (%)	3 (%)		
C7676-*bla_KPC_*	119.40	229.70	175.67	6	R
C5343-*bla_KPC_*	100.00	59.03	27.26	3	I
C7673-*bla_KPC_*	103.94	118.29	106.10	5	R
C7674-*bla_NDM_*	54.21	0.86	0.86	2	S
C7664-*bla_NDM_*	99.08	0.33	0.00	2	S
C7662-*bla_NDM_*	99.56	0.00	0.00	2	S
K2044	108.80	107.23	101.16	5	R
ATCC^®^13883	1.95	0.00	0.00	1	S

Serum killing results were shown by the percent survival of all isolates after incubation in normal human serum at one, two and three h. The grade and interpretive are classified as previously described [22]. R, resistance; S, susceptible; and I, intermediately sensitive.

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
