# Peer review of "ST11 Carbapenem-Resistant Klebsiella pneumoniae Clone Harboring blaNDM Replaced a blaKPC Clone in a Tertiary Hospital in China"

_antibiotics, 2022, doi:10.3390/antibiotics11101373_

Round 1

Reviewer 1 Report

Given the global problem of multiple resistance to antibiotics, molecular epidemiology studies are of great importance. This manuscript addresses very clearly, using state-of-the-art methodology, the serious situation of antimicrobial resistance experienced in hospitals.

In my opinion, the authors have carried out an important and excellent research work that reveals the seriousness of antimicrobial resistance towards last-generation antibiotics such as carbapanems.

Work on the molecular epidemiology of antimicrobial resistance, such as this one, is extremely useful for adopting containment measures to reduce this serious crisis

Author Response

Response: Thank you for your positive comments. We have re-edited the manuscript. Please see the clean version.

Reviewer 2 Report

Authors executed the study and analysed the data on a longitudinal basis for the change in resistance pattern in Enterobacteriales. Finding the relevance of the growth pattern of the newly developing resistant strains was an important highlight of the study. The results obtained support the research question. However, the following aspects can be improved in the manuscript. 

Inclusion of a couple of relevant citations from 2022. 

Add a few sentences at the end of the discussion which concludes the study.  

Author Response

Authors executed the study and analysed the data on a longitudinal basis for the change in resistance pattern in Enterobacteriales. Finding the relevance of the growth pattern of the newly developing resistant strains was an important highlight of the study. The results obtained support the research question. However, the following aspects can be improved in the manuscript.

Response: Thank you for your positive comments.

Inclusion of a couple of relevant citations from 2022.

Response: Thanks. We have added the latest relevant citations from 2022.

Add a few sentences at the end of the discussion which concludes the study.

Response: Thanks. We have added the conclusion in the end of the discussion. Please see the clean version. (Lines 374-377)

Reviewer 3 Report

The manuscript studies the prevalence of carbapenamase resistant Enterobacterales in China. This is a well-researched paper and the authors did a very good job in giving detailed explanation of the methods used in the study. Few comments for improvement is provided below.

1.       Please expand MHA in line 90.

2.       Please provide the source information of media used Line 88

3.       Please change "them" to "then" in line 96

4.       Figures caption should be self-explanatory or should explain everything. In Figure 2, the color does not indicate what they are meant for. Figure 4 and 5 does not say what the error bar represents ie., Standard error or SEM

5.       Please follow the reference style of the Antibiotics journal. Line 302

6.       Line 324 -326. “ In this study, CRE strains …..” These results were not presented anywhere in the results. Please include these results in the Results section as it is an important piece of information.

7.       Please include author contributions, conflicts of interest, Funding information. Line 375 -378

Author Response

The manuscript studies the prevalence of carbapenamase resistant Enterobacterales in China. This is a well-researched paper and the authors did a very good job in giving detailed explanation of the methods used in the study. Few comments for improvement is provided below.

Response: Thank you for your positive comments.

  1. Please expand MHA in line 90.

Responses: Thanks. It has been edited (Line 92).

  1. Please provide the source information of media used Line 88

Responses: Thank you, the source information of media has been added in the text. (Line 90).

  1. Please change "them" to "then" in line 96

Responses: Thanks. It has been edited.

  1. Figures caption should be self-explanatory or should explain everything. In Figure 2, the color does not indicate what they are meant for. Figure 4 and 5 does not say what the error bar represents ie., Standard error or SEM

Responses: We have revised the Figures caption to make it more self-explanatory. Please see the Figures caption in the clean version.

  1. Please follow the reference style of the Antibiotics journal. Line 302

Responses: It has been edited (Line 301).

  1. Line 324 -326. “In this study, CRE strains …..” These results were not presented anywhere in the results. Please include these results in the Results section as it is an important piece of information.

Responses: Thanks. We have added these description in the Results section. (Lines 224-226)

  1. Please include author contributions, conflicts of interest, Funding information. Line 375 -378

Responses: Thanks. We have added author contributions, conflicts of interest, and Funding information in the manuscript. (Lines 378-388)
